# Aligning Individual and Collective Objectives in Multi-Agent Cooperation

**Yang Li**
The University of Manchester
yang.li-4@manchester.ac.uk

**Wenhao Zhang**
Shanghai Jiao Tong University
wenhao_zhang@sjtu.edu.cn

**Jianhong Wang**[†]
INFORMED-AI Hub
University of Bristol
jianhong.wang@bristol.ac.uk

**Shao Zhang**
Shanghai Jiao Tong University
shaozhang@sjtu.edu.cn

**Yali Du**
King's College London
yali.du@kcl.ac.uk

**Ying Wen**[∗]
Shanghai Jiao Tong University
ying.wen@sjtu.edu.cn

**Wei Pan**[∗]
The University of Manchester
wei.pan@manchester.ac.uk

## Abstract

Among the research topics in multi-agent learning, mixed-motive cooperation is one of the most prominent challenges, primarily due to the mismatch between individual and collective goals. The cutting-edge research is focused on incorporating domain knowledge into rewards and introducing additional mechanisms to incentivize cooperation. However, these approaches often face shortcomings such as the effort on manual design and the absence of theoretical groundings. To close this gap, we model the mixed-motive game as a differentiable game for the ease of illuminating the learning dynamics towards cooperation. More detailed, we introduce a novel optimization method named *A*ltruistic *G*radient *A*djustment (*AgA*) that employs gradient adjustments to progressively align individual and collective objectives. Furthermore, we theoretically prove that AgA effectively attracts gradients to stable fixed points of the collective objective while considering individual interests, and we validate these claims with empirical evidence. We evaluate the effectiveness of our algorithm AgA through benchmark environments for testing mixed-motive collaboration with small-scale agents such as the two-player public good game and the sequential social dilemma games, Cleanup and Harvest, as well as our self-developed large-scale environment in the game StarCraft II.

## 1 Introduction

Multi-agent cooperation primarily focuses on learning how to promote collaborative behavior in shared environments. In general, multi-agent cooperation research is categorized into two prominent areas: pure-motive cooperation and mixed-motive cooperation [Du et al., 2023, McKee et al., 2020]. Recent progress in cooperative Multi-Agent Reinforcement Learning (MARL) has been primarily focusing on pure-motive cooperation, also known as common payoff games. This game models situations where each agent's individual goal fully aligns with the collective objectives [Yu et al., 2022, Wang et al., 2020, Zhong et al., 2024, Li et al., 2024]. Nevertheless, mixed-motive cooperation

---

[∗]Corresponding authors. [†] Jianhong Wang is a visiting researcher at the University of Manchester.

38th Conference on Neural Information Processing Systems (NeurIPS 2024).

is more widespread across real-world situations. It is usually defined by imperfect alignment between individual and collective rationalities [Rapoport, 1974, McKee et al., 2020].

Recent studies in mixed-motive cooperative MARL have largely employed hand-crafted designs to promote collaboration. One popular approach is to align objectives as per existing mechanisms in cooperative games, such as reputation [Anastassacos et al., 2021], norms [Vinitsky et al., 2023], and contracts [Hughes et al., 2020]. Another prevalent method leverages intrinsic motivation to align individual and collective objectives, enhancing altruistic collaboration by integrating heuristic knowledge into the incentive function. Conventionally, some works accumulate individual rewards with the group to promote altruistic conduct [Hostallero et al., 2020, Peysakhovich and Lerer, 2018, Apt and Schäfer, 2014, Roesch et al., 2024]. Furthermore, some studies derive more sophisticated preference signals from the rewards of other agents [Hughes et al., 2018, McKee et al., 2020]. Additionally, several approaches aim to learn the potential influences of an agent's actions on others [Yang et al., 2020, Jaques et al., 2019, Lu et al., 2022]. Most of these algorithms rely heavily on carefully crafted designs, necessitating significant human expertise and detailed domain knowledge. On the other hand, several studies leverage Nash equilibria and related game theory concepts, such as the price of anarchy, to automatically modify rewards by learning additional weights that adjust the original objectives [Gemp et al., 2022, Kwon et al., 2023]. However, finding Nash equilibria in nonconvex games presents a greater challenge compared to identifying minima in neural networks [Letcher et al., 2019a].

In this study, we introduce the differentiable mixed-motive game (DMG), an effective framework for analyzing learning dynamics at both individual and collective levels. Furthermore, we derive the Altruistic Gradient Adjustment (AgA) algorithm, which aligns individual and collective objectives by modifying the gradient. We theoretically prove that AgA, with an appropriately chosen sign for the adjustment term, can successfully guide the gradient towards stable fixed points of the collective objective while considering individual interests. Additionally, empirical evidence from optimization trajectory visualizations and ablation studies validates our claims.

We also conduct comprehensive experiments to verify the effectiveness of the proposed AgA algorithm. First, optimization trajectory analysis and ablation studies validate our theoretical conclusions. Next, a series of experiments conducted in various environments demonstrate that the AgA algorithm outperforms related baselines in both gradient adjustment and mixed-motive cooperation areas. In addition to commonly used testbeds like the public goods matrix game and sequential social dilemma games (Cleanup and Harvest) [Leibo et al., 2017], which are limited in terms of agent scale, action space, and task complexity, we introduce a more complex mixed-motive environment called Selfish-MMM2, an adaptation of the MMM2 map from the StarCraft II game [Samvelyan et al., 2019]. Selfish-MMM2 offers the following significant improvements over other mixed-motive games: it supports large-scale scenarios with 10 heterogeneous controlled agents facing 11 enemies, features a large action space with a size of $18^{10}$, which vastly exceeds the action spaces in Cleanup $9^5$ and Harvest $8^5$ that are limited to 5 homogeneous agents. Selfish-MMM2 also has a larger action space and greater task complexity than the 10-player PD testbed (with a size of $2^{10}$) used in D3C [Gemp et al., 2022], which claims to solve the large-scale problem in mixed-motive games.

In summary, the contributions of this paper are as follows: (1) We are the first work to model the mixed-motive game as a differentiable game (to the best of our knowledge) and propose the AgA algorithm to align individual and collective objectives from a gradient perspective. (2) We theoretically prove that, in the neighborhood of fixed points, AgA could pull the gradient toward stable fixed points of the collective objectives and push the gradient away from unstable fixed points. (3) We introduce Selfish-MMM2, a novel large-scale mixed-motive cooperation environment, and conduct comprehensive experiments across multiple settings that verify our theoretical claims and demonstrate the superior performance of the AgA algorithm.

## 2 Related Work

**Mixed-motive cooperation.** Mixed-motive cooperation refers to scenarios where the group's objectives are sometimes aligned and at other times conflicted [Philip S. Gallo and McClintock, 1965]. Recently, there has been a surge in academic interest in the Sequential Social Dilemma (SSD) [Leibo et al., 2017], which expands the concept from its roots in matrix games [Macy and Flache, 2002] to Markov games. Prosocial [Peysakhovich and Lerer, 2018] improves collective performance by

blending individual rewards to redraft the agent's overall utility. Inequity aversion further integrates the concept into Markov games by adding envy (disadvantageous inequality) and guilt (advantageous inequality) rewards to the original individual rewards [Hughes et al., 2018]. To further promote cooperation, the gifting mechanism—a crucial strategy in mixed-motive cooperation [Lupu and Precup, 2020]—allows agents to influence each other's reward functions through peer rewarding. Besides, PED-DQN [Hostallero et al., 2020] introduces an automatic reward-shaping MARL method that gradually adjusts rewards to shift agents' actions from their perceived equilibrium towards more cooperative outcomes. Social Value Orientation (SVO) strategy introduces a unique shared rewards-based compensation approach, encouraging behavior modifications in line with interdependence theory [McKee et al., 2020]. The LIO strategy bypasses the need to modify extrinsic rewards by empowering an agent to directly influence its partner's actions [Yang et al., 2020]. Meanwhile, other studies introduce new cooperative mechanisms such as incorporating a reputation model [McKee et al., 2021], social norms [Vinitsky et al., 2023]. Recently, Roesch et al. [2024] proposed a selfishness level method that incorporates social welfare into individual rewards to enhance altruistic cooperation. Additionally, several works have explored automatically modifying rewards online. D3C [Gemp et al., 2022] learns to mix rewards to improve efficiency in a Nash equilibrium. Kwon et al. [2023] extended D3C by addressing the problem of automatically modifying individual agent objectives to optimize a desired global objective. Despite these advances, many current studies lack both cost efficiency in design and theoretical analysis of alignment and convergence. To address this, our study applies a gradient perspective to align goals and further investigates the learning dynamics of our proposed method.

**Gradient-based Methods.** Our proposed AgA is fundamentally a gradient adjustment methodology, making gradient-based optimization methods highly relevant to the context of this paper. Methods based on gradient have been developed to find stationary points, such as the Nash equilibrium or stable fixed points. Optimistic Mirror Descent leverages historical data to extrapolate subsequent gradients Daskalakis et al. [2018], while Gidel et al. [2019] extends this concept by advocating averaging methodologies and variants of extrapolation techniques. Consensus gradient adjustment, or consensus optimization, is a technique that embeds a consensus agreement term within the gradient to ensure its convergence [Mescheder et al., 2017]. Learning with Opponent-Learning Awareness (LOLA) utilizes information from other players to compute one player's anticipated learning steps [Foerster et al., 2018]. Subsequently, Stable Opponent Shaping (SOS) [Letcher et al., 2019b] and Consistent Opponent-Learning Awareness (COLA) [Willi et al., 2022] methods have enhanced the LOLA algorithm, targeting convergence assurance and inconsistency elimination, respectively. Symplectic Gradient Adjustment (SGA) alters the update direction towards the stable fixed points based on a novel decomposition of game dynamics [Balduzzi et al., 2018, Letcher et al., 2019a]. Recently, Learning to Play Games (L2PG) ensures convergence towards a stable fixed point by predicting updates to players' parameters derived from historical trajectories [Chen et al., 2023]. However, these methods, while focusing on zero-sum or general sum games, could potentially act counter to their individual interests, as they may prioritize stability over minimizing personal loss. Our research specifically targets mixed-motive setting with the aim of reconciling individual and collective objectives.

## 3 Preliminaries

### 3.1 Differential Game

The theory of differential games was initially proposed by Isaacs [1965], aiming to expand the scope of sequential game theory to encompass continuous-time scenarios. Through the lens of machine learning, we formalize the differential game, as shown in Definition 3.1.

**Definition 3.1** (Differential Game [Balduzzi et al., 2018, Letcher et al., 2019a])**.** *A differential game could be defined as a tuple $\{\mathcal{N}, \boldsymbol{w}, \boldsymbol{\ell}\}$, where $\mathcal{N} = \{1, \ldots, n\}$ is the set of players. The parameter set $\boldsymbol{w} = [\boldsymbol{w}_i]^n \in \mathbb{R}^d$ is defined, each with $\boldsymbol{w}_i \in \mathbb{R}^{d_i}$ and $d = \sum_{i=1}^n d_i$. Here, $\boldsymbol{\ell} = \{\ell_i : \mathbb{R}^d \to \mathbb{R}\}_{i=1}^n$ represents the corresponding losses. These losses are assumed to be at least twice differential. Each player $i \in \mathcal{N}$ is equipped with a policy, parameterized by $\boldsymbol{w}_i$, aiming to minimize its loss $\ell_i$.*

We write the *simultaneous gradient* $\boldsymbol{\xi}(\boldsymbol{w})$ of a differential game as $\boldsymbol{\xi}(\boldsymbol{w}) = (\nabla_{\boldsymbol{w}_1}\ell_1, \ldots, \nabla_{\boldsymbol{w}_n}\ell_n) \in \mathbb{R}^d$, which is the gradient of the losses with respect to the parameters of the respective players. Furthermore, *Hessian matrix $\boldsymbol{H}$* mentioned in a differential game is the Jacobian matrix of the simultaneous gradient.

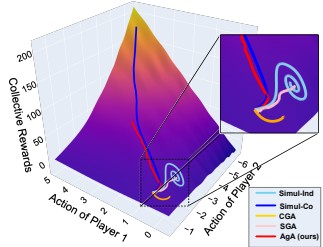
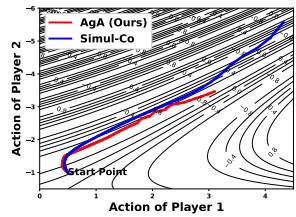
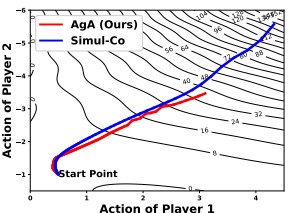

(a) Collective Reward Landscape     (b) Reward Contour of Player 1     (c) Reward Contour of Player 2

Figure 1: Trajectories of optimization in a two-player DMG (as delineated in Example 4.1). Fig.1a displays the trajectories over the collective reward landscape - deeper orange equates to higher rewards. Remarkably, only Simul-Co and AgA make successful strides towards the social optimum. Fig.1b and Fig. 1c delineate trajectories on the individual reward contour, *underscoring Simul-Co's neglect for Player 1's interests as it navigates through the crests and troughs of its reward.* Conversely, our AgA optimizes along the summit of Player 1's reward while also maximizing the collective reward, *demonstrating successful alignment.*

The *learning dynamics* of differential game often refers to the process of sequential updates over $\boldsymbol{w}$. The *learning rule* for each player is defined as the operator, $\boldsymbol{w} \leftarrow \boldsymbol{w} - \gamma \boldsymbol{\xi}$, where $\gamma$ is a step size (learning rate) to determine the distance to move for each update.

## 3.2 Gradient Adjustment Optimization

The *stable fixed point*, a criterion initiated from stability theory, exhibits its stability (robustness) to minor perturbations of environments, making it applicable to many real-world scenarios.

**Definition 3.2.** *A point $\boldsymbol{w}^\star$ is a fixed point if $\boldsymbol{\xi}(\boldsymbol{w}^\star) = 0$. If $\boldsymbol{H}(\boldsymbol{w}^\star) \succeq 0$ and $\boldsymbol{H}(\boldsymbol{w}^\star)$ is invertible, the fixed point $\boldsymbol{w}^\star$ is called stable fixed point. If $\boldsymbol{H}(\boldsymbol{w}^\star) \prec 0$, the point is called unstable.*

A naive idea to steer the dynamic towards convergence at fixed points involves minimizing $\frac{1}{2}\|\boldsymbol{\xi}(\boldsymbol{w})\|^2$. Assuming the Hessian $\boldsymbol{H}(\boldsymbol{w})$ is invertible, then $\nabla(\frac{1}{2}\|\boldsymbol{\xi}(\boldsymbol{w})\|^2) = \boldsymbol{H}^T \boldsymbol{\xi} = 0$ holds true if and only if $\boldsymbol{\xi} = 0$. However, it could converge to unstable fixed points [Mescheder et al., 2017]. Hence, the consensus optimization method has been proposed, incorporating gradient adjustment [Mescheder et al., 2017], shown as follows: $\tilde{\boldsymbol{\xi}} = \boldsymbol{\xi} + \lambda \cdot \nabla \frac{1}{2}\|\boldsymbol{\xi}(\boldsymbol{w})\|^2 = \boldsymbol{\xi} + \lambda \cdot \boldsymbol{H}^T \boldsymbol{\xi}$. For simplicity, we will refer to the *consensus gradient adjustment* as *CGA*. While CGA proves effective in certain specific scenarios, such as two-player zero-sum games, it unfortunately falls short in general games [Balduzzi et al., 2018]. To address this shortage, *symplectic gradient adjustment* (*SGA*) [Balduzzi et al., 2018, Letcher et al., 2019a] is introduced to find the stable fixed point in general sum games, such that $\tilde{\boldsymbol{\xi}} = \boldsymbol{\xi} + \lambda \cdot \boldsymbol{A}^T \boldsymbol{\xi}$. Herein, $\boldsymbol{A}$ represents the antisymmetric component of the generalized Helmholtz decomposition of $\boldsymbol{H}$, $\boldsymbol{H}(\boldsymbol{w}) = \boldsymbol{S}(\boldsymbol{w}) + \boldsymbol{A}(\boldsymbol{w})$, where $\boldsymbol{S}(\boldsymbol{w})$ denotes the symmetric component.

## 4 Method

In this section, we first define differentiable mixed-motive games (DMG) and analyzes the alignment dilemma faced by existing methods, which is described in Section 4.1. We then propose the Altruistic Gradient Adjustment (AgA) algorithm in Section 4.2 to align the individual and collective objectives by modifying the gradient. Finally, a case study demonstrating AgA's effectiveness in a toy two-player DMG is presented in Section 4.3.

## 4.1 Differentiable Mixed-motive Game

We first formulate the mixed-motive game as a differentiable game. Specifically, the ***differentiable mixed-motive game (DMG)*** is defined as a tuple $(\mathcal{N}, \boldsymbol{w}, \boldsymbol{\ell})$, where $\ell_i \in \boldsymbol{\ell}$ is at least twice the differentiable loss function for the agent $i$. Differentiable losses exhibit the *mixed motivation property*: minimization of individual losses can result in a conflict between individuals or between individual and collective objectives (e.g., maximizing individual stats and winning the game are often

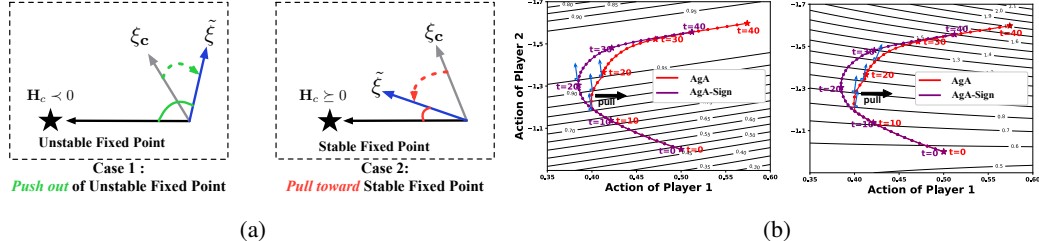

(a)                                                                                    (b)

Figure 2: **Left figure a: Illustration of Corollary 4.3.** In case 1, within an unstable fixed point's neighborhoods, an appropriate selection of the $\lambda$ sign push AgA to evade the unstable fixed point and pull towards a stable fixed point in its neighborhoods as shown in case 2. **Right figure b: Alignment Effectiveness of AgA.** The comparison between AgA (shown in red) and AgA without sign alignment (AgA-Sign, in purple) trajectories spans 40 steps, marked at every tenth step. Norm gradients are represented with blue arrows. Starting from the 14th step, sign alignment pulls the gradient toward the steepest direction, resulting in AgA reducing the number of steps by approximately 15% compared to AgA-Sign by the end of the trajectory.

conflict in basketball matches). In addition to the simultaneous gradient $\boldsymbol{\xi}(\boldsymbol{w})$ of individual losses with respect to the parameters of the respective players, we use $\boldsymbol{\xi}_c(\boldsymbol{w})$ to refer to the gradient of the collective loss: $\boldsymbol{\xi}_c(\boldsymbol{w}) = (\nabla_{\boldsymbol{w}_1}\ell_c, \ldots, \nabla_{\boldsymbol{w}_n}\ell_c)$.

**Alignment Dilemma in DMG.** Direct optimization of individual losses and collective loss are straightforward ideas to solving DMG problems. However, minimizing only the individual loss of each agent is unlikely to achieve a collective optimum [Leibo et al., 2017, McKee et al., 2020]. Conversely, optimizing the collective loss might produce better overall outcomes but risks neglecting individual interests [Roesch et al., 2024]. Additionally, the local convergence of gradient descent in singular collective functions is not always guaranteed [Balduzzi et al., 2018].

Example 4.1 gives a two-player differentiable mixed-motive game. We provide a visual representation of optimization trajectories using a series of methods to investigate the learning dynamics involved in resolving the example, as shown in Fig. 1.

**Example 4.1.** *Consider a two-player DMG with $\ell_1(a_1, a_2) = -sin(a_1 a_2 + a_2^2)$ and $\ell_2(a_1, a_2) = -[cos(1 + a_1 - (1 + a_2)^2) + a_1 a_2^2]$, where $a_i$ represents the action of the player $i$ ($i = 1, 2$), and $a_i \in \mathbb{R}$. The rewards for the two players are the negation of their respective losses.*

We first implement simultaneous optimization of individual losses (Simul-Ind) with respect to each parameter, defined by the learning rule $\boldsymbol{w}_i = \boldsymbol{w}_i - \gamma\boldsymbol{\xi}_i$, for $i \in \{1, 2\}$, where $\boldsymbol{\xi}_i = \nabla_{\boldsymbol{w}_i}\ell_i$. Simultaneous optimization of collective loss (Simul-Co) replaces the individual gradient with the collective gradient $\boldsymbol{\xi}_c$ of collective loss $\ell_c = \ell_1 + \ell_2$. Furthermore, in order to investigate the learning dynamics of prevalent gradient modification optimization approaches such as SGA and CGA for the two-player differentiable mixed-motive game, we modify the learning rule by integrating the respective adjusted gradient as delineated in Section 3.2.

Fig. 1a shows the optimization paths on the collective reward landscapes, with every path starting from the front-bottom of the landscape. The collective reward is defined as social welfare, i.e., the sum of individual rewards. As shown in the figure, Simul-Ind, CGA , and SGA converge to unstable points or local maxima. Simul-Co effectively navigates towards the apex of the collective reward landscape as depicted in Fig. 1a. However, Simul-Co is ineffective in aligning individual and collective objectives, leading to the neglect of individual interests. As depicted in Fig. 1b, the update trajectory navigates through the crests and troughs of Player 1's reward contour. The trajectory suggests a disregard for Player 1's preferences, signifying that the updates are predominantly driven by the overarching collective goal.

## 4.2 Altruistic Gradient Adjustment

To tackle the challenge of aligning objectives in DMG, we propose the Altruistic Gradient Adjustment (AgA) method. Unlike existing gradient adjustment techniques [Mescheder et al., 2017, Balduzzi et al., 2018, Letcher et al., 2019a, Chen et al., 2023], which primarily aim at achieving stable fixed

points for individual objectives, our AgA method simultaneously considers both individual and collective objectives when seeking stable fixed points for the collective objective. Specifically, AgA is defined as follows.

**Definition 4.2** (Altruistic Gradient Adjustment). *Altruistic gradient adjustment (AgA) extends the gradient term in the learning dynamic as*

$$\tilde{\boldsymbol{\xi}} := \boldsymbol{\xi}_c + \lambda \boldsymbol{\xi}_{adj} = \boldsymbol{\xi}_c + \lambda \left( \boldsymbol{\xi} + \boldsymbol{H}_c^T \boldsymbol{\xi}_c \right), \tag{1}$$

*where $\lambda \in \mathbb{R}$ is alignment parameter, $\lambda \boldsymbol{\xi}_{adj}$ is called adjustment term. In $\boldsymbol{\xi}_{adj}$, $\boldsymbol{\xi}_c$ and $\boldsymbol{H}_c$ is the gradient vector and Hessian matrix of the game about collective loss.*

Note that the Hessian matrix $\boldsymbol{H}_c$ is symmetric. For brevity, we denote $\nabla \mathcal{H}_c$ as $\nabla_{\boldsymbol{w}} \left( \frac{1}{2} \|\boldsymbol{\xi}_c\|^2 \right)$, which simplifies to $\boldsymbol{H}_c^T \boldsymbol{\xi}_c$. In our AgA method, the formula $\xi_c + \lambda(\xi + \boldsymbol{H}_c^T \xi_c)$ includes the Hessian matrix, but we do not compute it directly. Instead, we calculate Hessian-vector products $\boldsymbol{H}_c^T \xi_c$, which suffice for determining the adjustment gradient and the sign of $\lambda$. This approach reduces computational complexity, enhancing the effectiveness of the AgA method. **The cost for computing Hessian-vector products $\boldsymbol{H}_c^T \xi_c$ is $\mathcal{O}(n)$ for $n$ weights** [Pearlmutter, 1994]. Refer to Section 6 for a detailed analysis.

While AgA introduces additional complexity, we theoretically demonstrate that an appropriate choice of the sign of $\lambda$ ensures that AgA pulls the gradient towards a stable fixed point through the adjustment term in the neighborhood of fixed points. Conversely, when dealing with an unstable fixed point, AgA will push the gradient away from these unstable points.

Before introducing the corollary, we first provide some basic notations. The inner product of vectors $\boldsymbol{a}$ and $\boldsymbol{b}$ are denoted by $\langle \boldsymbol{a}, \boldsymbol{b} \rangle$. If the Hessian matrix $\boldsymbol{H}$ is non-negative-definite, $\langle \boldsymbol{\xi}_c, \nabla \mathcal{H} \rangle \geq 0$ for a non-zero $\boldsymbol{\xi}_c$. Analogously, if $\boldsymbol{H}$ is negative-definite, $\langle \boldsymbol{\xi}_c, \nabla \mathcal{H} \rangle < 0$ for a non-zero $\boldsymbol{\xi}_c$. Lastly, the angle between the two vectors $\boldsymbol{a}$ and $\boldsymbol{b}$ is denoted by $\theta(\boldsymbol{a}, \boldsymbol{b})$. Then, we state the corollary as follows:

**Corollary 4.3.** *In the neighborhood of fixed points of the collective objective, AgA will **pull** the gradient **toward** stable fixed points, which means $\theta(\tilde{\boldsymbol{\xi}}, \nabla \mathcal{H}_c) \leq \theta(\boldsymbol{\xi}_c, \nabla \mathcal{H}_c)$, and **push away** from unstable ones, indicated by $\theta(\tilde{\boldsymbol{\xi}}, \nabla \mathcal{H}_c) \geq \theta(\boldsymbol{\xi}_c, \nabla \mathcal{H}_c)$, if $\lambda$ satisfies $\lambda \cdot \langle \boldsymbol{\xi}_c, \nabla \mathcal{H}_c \rangle (\langle \boldsymbol{\xi}, \nabla \mathcal{H}_c \rangle + \|\nabla \mathcal{H}_c\|^2) \geq 0$.*

The proof is provided in Appendix B. Fig. 2a presents an illustrative visualization of Corollary 4.3. The figure depicts two scenarios: the neighborhoods of stable and unstable fixed point (denoted by a star), respectively. In the neighborhood of an unstable fixed point of the collective objective, as depicted in Case 1, our AaA gradient $\tilde{\boldsymbol{\xi}}$ is pushed out of the region, resulting in a faster escape compared to the original collective gradient $\boldsymbol{\xi}_c$. This behavior is exemplified by the green arrow. Conversely, Case 2 illustrates the scenario of a stable fixed point, wherein our AaA gradient $\tilde{\boldsymbol{\xi}}$ is pulled towards the stable equilibrium, exemplified by the red arrow.

AgA could be easily integrated into any centralized training with decentralized execution (CTDE) framework within MARL. Detailed statements on the implementation of AgA, including the pseudocode, are provided in Appendix C.

## 4.3 Alignment Effectiveness of AgA: A Toy Experiment

To address the two-player DMG as outlined in Example 4.1, we incorporate the AgA method into the fundamental gradient descent algorithm. The optimization trajectories in both the collective reward landscape and the individual player reward contour are represented in Fig. 1a, Fig. 1b, and Fig. 1c. In contrast to the objective misalignment exhibited by Simul-Co, AgA successfully aligns the agents' interests, as evidenced in Fig. 1b and Fig 1c. Fig. 1b depicts the trajectory of AgA meticulously carving its course along the summit of the individual reward contour. Furthermore, AgA is also shown to improve Player 2 reward (as shown in Fig. 1c), despite a slower convergence rate than Simul-Co.

In addition, Fig. 2b presents a critical comparison between the trajectories of AgA (in red) and AgA without sign alignment (AgA-Sign, depicted in purple), as outlined in Corollary 4.3. This side-by-side comparison covers 40 steps and features highlighted points every tenth step. It provides a visual illustration of the effectiveness introduced by sign alignment starting from the 14th step in the trajectories of AgA and AgA without sign alignment. Remarkably, norm gradients are represented by

Table 1: The comparison of the average individual rewards (denoted as $r_1, r_2$), social welfare (denoted as $SW$), and equality metric (denoted as $E$) on two- player public goods game. We show the mean of value and 95% confidence interval utilizing 50 random runs.

| Metrics | Simul-Ind | CGA | SGA | SVO | Simul-Co | SL | AgA |
|---|---|---|---|---|---|---|---|
| $r_1$ | $1.133 \pm 0.063$ | $1.156 \pm 0.060$ | $1.175 \pm 0.062$ | $1.104 \pm 0.054$ | $1.433 \pm 0.056$ | $1.314 \pm 0.062$ | $\mathbf{1.443\pm 0.042}$ |
| $r_2$ | $1.184 \pm 0.065$ | $1.150 \pm 0.057$ | $1.137 \pm 0.063$ | $1.060 \pm 0.051$ | $1.381 \pm 0.065$ | $1.371 \pm 0.057$ | $\mathbf{1.459 \pm 0.041}$ |
| SW | $2.316\pm 0.039$ | $2.306\pm 0.039$ | $2.312\pm 0.044$ | $2.164 \pm 0.026$ | $2.814 \pm 0.033$ | $2.684 \pm 0.049$ | $\mathbf{2.903\pm 0.023}$ |
| E | $0.923\pm 0.014$ | $0.929\pm 0.012$ | $0.924\pm 0.013$ | $0.930 \pm 0.011$ | $0.941 \pm 0.014$ | $0.940 \pm 0.011$ | $\mathbf{0.960 \pm 0.008}$ |

blue arrows, indicating the direction of the fastest updates. As depicted in the figure, AgA with sign selection is aligned toward the steepest update direction, resulting in a reduction of approximately 15% in the number of steps compared to AgA-Sign by the end of the trajectory.

## 5 Experiments

To assess the effectiveness of the proposed AgA algorithm, we perform comprehensive experiments across various environments, ranging from simple to complex, and from small to large scales. The initial environment involves a two-player public goods game (see Section 5.1), followed by sequential social dilemma environments (see Section 5.2): Cleanup and Harvest, involving 5 homogeneous players. The final test is conducted in our specially developed selfish-MMM2 environment (see Section 5.3), which is both more complex and larger in scale, featuring approximately 10 controlled agents of 3 distinct types, competing against 11 adversaries. Comparative experiments employ commonly used baseline approaches, such as simultaneous optimization with individual and collective losses (termed Simul-Ind and Siml-Co), CGA [Mescheder et al., 2017], SGA [Balduzzi et al., 2018], SVO [McKee et al., 2020], and the selfishness level approach (termed SL) [Roesch et al., 2024].

### 5.1 Two-Player Public Goods Game

A two-player public goods matrix game $\mathcal{G}$ is widely utilized to study cooperation in social dilemmas. The game involves players $\{1, 2\}$ with parameters $\{\boldsymbol{w}_1, \boldsymbol{w}_2\}$ and payoffs $\{p_1, p_2\}$. The social welfare, denoted as $SW = p_1 + p_2$. Each player, i.e., $i \in \{1, 2\}$, contributes an amount $a_i$ within a budgeted range $[0, b]$, and the host evenly distributes these contributions as $\frac{c}{2}(a_1 + a_2)$, where $1 < c \leq 2$. Consequently, each player's payoff $p_i(a_1, a_2)$ is estimated as $b - a_i + \frac{c}{2}(a_1 + a_2)$. In our experiments, we set the budget $b$ to 1 and weight $c$ to 1.5, with the social optimum of the game at $(1, 1)$.

**Results.** Table 1 presents a comparison of individual rewards and the collective outcome, derived from 50 random trials, each limited to 100 steps. The rows $r_1$, $r_2$, $SW$, and $E$ represent the individual rewards for each player, the group's total welfare (SW), and the equality metric (E), respectively. The equality metric is based on the Gini coefficient ($G$) [David, 1968], with $E$ defined as $E = 1 - G = 1 - \frac{2}{n^2 \bar{p}} \sum_{i=1}^{n} i(p_i - \bar{p})$, where $\bar{p}$ is the mean of the ranked payoff vector $p$, and $n$ is the total number of players. A higher $E$ value indicates greater equality among the players.

The social optimum of the game occurs when all players allocate their entire budget, achieving the highest possible social welfare score of 3. Among all algorithms, AgA achieves the closest social welfare score to this optimum, with a value of 2.90. Furthermore, AgA exhibits the greatest degree of fairness, with two players receiving nearly identical rewards and the highest equality value, in contrast to the baseline algorithms. Furthermore, Fig. 4 in Appendix D illustrates the action distributions of these algorithms, showing that AgA actions are more tightly clustered around the social optimum actions $(1, 1)$. This further indicates that AgA approaches the social optimum more effectively.

### 5.2 Sequential Social Dilemma: Cleanup and Harvest

We then verify the AgA algorithm in two widely used sequential social dilemma games with 5 homogeneous agents: Harvest and Cleanup [Hughes et al., 2018]. In the Cleanup scenario, agents predominantly gather rewards by harvesting apples in an orchard, where the yield is affected by the river's pollution levels. Neglecting rising pollution levels ultimately ceases apple production, thus setting up a trade-off between individual gains and communal welfare. In contrast, the Harvest

Table 2: The comparison of the average equality metric (denoted as $E$) on Harvest and Cleanup. We show the mean of equality value and standard deviation utilizing three random runs.

| Envs | Simul-Ind | Simul-Co | SVO | CGA | SL | AgA | | | |
|---|---|---|---|---|---|---|---|---|---|
| | | | | | | $\lambda = 0.1$ | $\lambda = 1$ | $\lambda = 100$ | $\lambda = 1000$ |
| **Harvest** | 0.973 $\pm$ 0.005 | 0.975 $\pm$ 0.006 | 0.974 $\pm$ 0.007 | 0.950 $\pm$ 0.051 | 0.972 $\pm$ 0.005 | 0.981 $\pm$ 0.006 | **0.988** $\pm$ **0.003** | 0.982 $\pm$ 0.012 | 0.980 $\pm$ 0.006 |
| **Cleanup** | 0.841 $\pm$ 0.071 | 0.948 $\pm$ 0.013 | 0.902 $\pm$ 0.019 | 0.903 $\pm$ 0.034 | 0.946 $\pm$ 0.016 | 0.940 $\pm$ 0.017 | 0.956 $\pm$ 0.007 | **0.959** $\pm$ **0.011** | 0.905 $\pm$ 0.022 |

scenario compensates agents for apple collection, with the regeneration of apples being ideally dependent on the proximity to other apples. Here, the communal challenge manifests itself as over-harvesting apples, which diminishes their regrowth rate and, consequently, the aggregate rewards for the group. Our research assessed AgA against various benchmarks, including Simul-Ind, Simul-Co, CGA, SVO, and SL. Simul-Ind was implemented using the IPPO algorithm [de Witt et al., 2020], while the remaining benchmarks employed the common parameters of the IPPO framework. The formula for calculating the collective loss for CGA and AgAs, which aims to improve both the performance of the group and the equitable distribution between agents, is expressed as $\sum_i \left( r_i - \alpha(1 - \arctan\left(\frac{\sum_{j,j \neq i} r_j}{r_i}\right)\right)$, where $\alpha$ is a constant. Further details on the experimental setups and the algorithms used are provided in Appendix E.1.

**Varying $\lambda$ values.** Fig. 3a and Fig. 3b explore the social welfare outcomes from AgA under three distinct $\lambda$ values: 0.1, 1, 100, and 1000. Among these, $\lambda = 100$ demonstrates superior performance in both the Cleanup and Harvest scenarios. Therefore, in the subsequent experiments involving sign alignment and comparing primary results with baseline models, we will employ $\lambda = 100$ as the default parameter.

**The effectiveness of sign alignment.** In the second row of Fig. 3, Fig. 3d and Fig. 3e illustrate the social welfare comparisons between AgA and AgA without sign alignment (AgA-Sign) during their training phases. Referencing Corollary 4.3, near a fixed point, sign alignment helps direct the gradient towards stable fixed points and away from unstable ones. During the latter half of the training sessions shown in Fig. 3d and Fig. 3e, it is observed that the social welfare ($SW$) for AgA-Sign does not improve to the same extent as it does for AgA, highlighting the effectiveness of sign alignment.

**Baseline Comparison.** As illustrated in Fig. 3g and Fig. 3h, AgA demonstrates a notable improvement in social welfare over Simul-Ind, Simul-Co, CGA, SVO, and SL. Particularly, within the Cleanup scenario, AgA recorded an average social welfare of 105.15, marking approximately a 56% increase over the runner-up method SL. In the Harvest scenario, AgA shows an average improvement of 33.55 in social welfare over the next best method SVO. Table 2 compares the mean and standard deviation of the equality metric achieved by different methods in the Harvest and Cleanup environments. A value closer to 1 signifies more equal rewards among agents. As shown in the table, AgA outperforms the baseline methods, demonstrating its ability to effectively balance the interests of all team members.

## 5.3 Selfish-MMM2

While sequential social dilemma games are frequently used to study mixed-motive problems, they are relatively simplistic compared to other experimental environments in MARL, particularly when considering real-world applications. To address this gap, we introduce a sophisticated mixed-motive testbed called Selfish-MMM2, which is adapted from the MMM2 map in the StarCraft II game [Samvelyan et al., 2019]. Unlike the standard SMAC framework, which employs a shared reward structure, our methodology assigns individual objectives to each agent, thereby enhancing the complexity of mixed motives. Specifically, we have redesigned the reward systems so that each agent's personal gain is directly tied to the damage they inflict on adversaries, diverging from traditional approaches where collective damage is evenly distributed among all agents. Additionally, penalties for agent elimination emphasize the importance of self-preservation and highlight agents' tendencies towards self-interested behavior. In comparison to widely utilized multi-agent environments such as Cleanup [Hughes et al., 2018], Harvest [Hughes et al., 2018], and the 10-player Prisoner's Dilemma (PD) for large-scale evaluations [Gemp et al., 2022], Selfish-MMM2 is distinguished by two notable

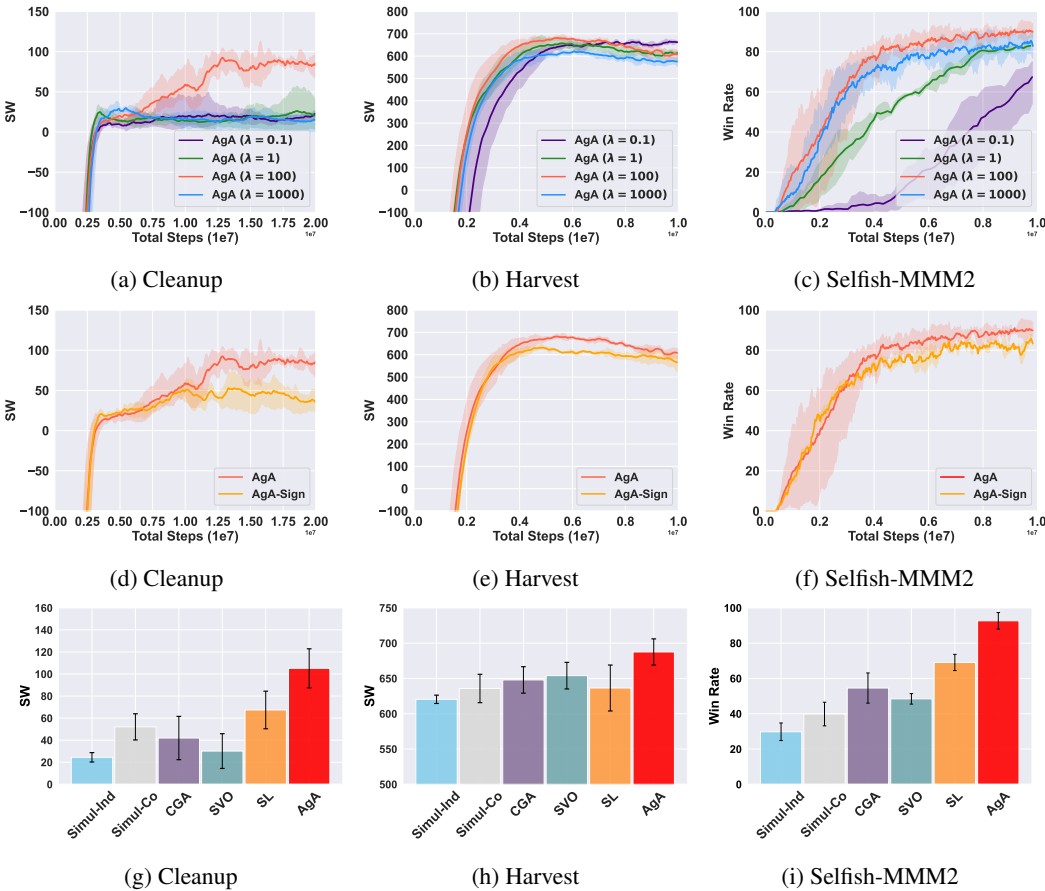

Figure 3: **The first row** displays results comparing different values of the alignment parameter $\lambda$ across three environments: Cleanup and Harvest (measurement of social welfare, SW) and Selfish-MMM2 (focusing on win rate). **The second row** examines the performance differences between the proposed AgA and AgA without sign alignment (AgA-Sign) on the three testbeds. The bold lines indicate the mean social welfare calculated in three seeds, while the surrounding shaded areas represent the 95% confidence interval. **The third row** compares the AgA method with baseline approaches on these testbeds. Each bar represents the mean collective results of each method and the error bars indicate the 95% confidence interval.

features: 1) Large-Scale Agents: Selfish-MMM2 manages the interactions of 10 heterogeneous agents, categorized into three types: seven Marines, two Marauders, and one Medivac, confronting 11 enemy units. This configuration is in stark contrast to Cleanup and Harvest, which each utilize five homogeneous agents. 2) Extensive Action Space: The action space in Selfish-MMM2, sized at $18^{10}$, is vastly larger than that of Cleanup and Harvest, which are limited to $9^5$ and $8^5$, respectively. It also surpasses the action space of the 10-player PD, which is $2^{10}$.

In evaluating the Selfish-MMM2 environment, we use the average win rate against the built-in AI bot in SMAC to measure the collective performance. The experimental setup employs the IPPO algorithm [de Witt et al., 2020] for the Simul-Ind algorithm, while the MAPPO [Yu et al., 2022] is utilized for the Simul-Co, SVO, CGA, SL and AgA algorithms. More details on the Selfish-MMM2 environment and implementation specifics are available in the Appendix E.2.

**Varying $\lambda$ Values and the effectiveness of Sign Alignment:** Fig. 3c presents the average win rates in various settings of the align parameter $\lambda$ in the AgA scenario. The findings indicate that increased $\lambda$ values are associated with faster convergence in the training period. Among these, a $\lambda$ value of 100 demonstrates a higher overall effectiveness compared to other values tested. Based on these findings, we have set a default value $\lambda$ of 100 for subsequent experiments. Furthermore, Fig. 3f in the second row illustrates a comparable conclusion in sequential social dilemma games. During the

Table 3: Comparison of the average running time between baseline methods and AgA in the two-player public goods game, including total duration, timesteps, time per step, and the time per step ratio relative to Simul-Ind.

| Metrics | Simul-Ind | Simul-Co | SL | CGA | SGA | AgA |
|---|---|---|---|---|---|---|
| Duration (ms) | 1165.79 | 910.15 | 1149.97 | 3041.84 | 3007.77 | 1034.69 |
| Steps | 4272 | 3252 | 3887 | 4478 | 4179 | 1389 |
| Step Time (ms) | 0.27 | 0.28 | 0.30 | 0.68 | 0.72 | 0.74 |
| Ratio | 1.00 | 1.04 | 1.11 | 2.52 | 2.67 | 2.74 |

initial training phase, the convergence rates of AgA and AgA-Sign are nearly identical. However, as training progresses, AgA exhibits convergence at points of superior performance, highlighting the critical role of sign alignment in achieving convergence near fixed points.

**Baseline Comparison.** As depicted in Fig. 3i, the bar chart displays a comparison of the average win rates along with 95% confidence intervals for AgA versus other baseline approaches. Our AgA approach attains the highest team performance, securing a 92.64% win rate with the built-in bot, which is 23.61 percent more than the next best win rate achieved by the SL method.

# 6 Discussion

**Computational Cost of AgA.** The altruistic gradient adjustment is expressed as $\tilde{\boldsymbol{\xi}} = \boldsymbol{\xi}_c + \lambda(\boldsymbol{\xi} + \boldsymbol{H}_c^T \boldsymbol{\xi}_c)$, where the key computational cost comes from the Hessian-vector product, $\boldsymbol{H}_c^T \boldsymbol{\xi}_c$. The cost for computing Hessian-vector products, $\boldsymbol{H}_c^T \xi_c$, is $\mathcal{O}(n)$ for $n$ weights [Pearlmutter, 1994]. This introduces added complexity compared to standard methods in mixed-motive MARL, making AgA's running time generally more than twice as long as gradient-based methods. Table 3 provides a comparison of the average running time between AgA and baseline methods in a two-player public goods game, including total duration, timesteps, time per step, and the time ratio relative to Simul-Ind. Simul-Ind, Simul-Co, and SL are standard gradient-based methods, while CGA and SGA modify gradients. Our results show that AgA takes 2-3 times longer per step but remains the most efficient, requiring only 1389 steps over 50 runs, compared to around 4000 steps for the baselines.

**Limitations.** Despite conducting a series of experiments to verify the proposed AgA algorithm, including using a new large-scale mixed-motive cooperation testbed, these environments remain somewhat removed from real-world applications. Although we strive to align the objectives of both individuals and the collective, our primary focus in this paper is on converging towards stable fixed points of the collective objective. For future research, we plan to delve deeper into the interaction of individual objectives in mixed-motive games, with an increased emphasis on understanding their dynamics in real-world mixed-motive cooperation scenarios.

# 7 Conclusion

In this paper, we propose AgA, an optimization method specifically designed to align individual and collective objectives through gradient adjustments in mixed-motive cooperation scenarios. To achieve this, we first model the mixed-motive game as a differentiable game, offering a powerful tool for analyzing learning dynamics at both individual and collective levels. Furthermore, we theoretically demonstrate that AgA can effectively attract gradients to stable fixed points and support our claims with empirical evidence. To evaluate the effectiveness of AgA in complex and large-scale scenarios, we introduce a new mixed-motive environment called Selfish-MMM2, which features heterogeneous large-scale agents, a larger action space, and increased task complexity. Comprehensive experiments, ranging from simple public goods games to Harvest, Cleanup, and Selfish-MMM2, show AgA's superior performance, consistently outperforming existing baselines across multiple evaluation metrics. These results validate our theoretical claims and highlight the effectiveness of AgA in achieving cooperative behavior in multi-agent systems.

## Acknowledgement

This work is partially supported by National Key R&D Program of China (2022ZD0114804) and National Natural Science Foundation of China (62106141). Jianhong Wang is fully supported by the Engineering and Physical Sciences Research Council [Grant Ref: EP/Y028732/1]. The authors also thank Shuqing Shi for his kind assistance and advice.

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

# A Broader Impacts

This paper presents a novel approach in the field of multi-agent learning, aimed at enhancing collaboration among agents with mixed motivations. The proposed framework addresses a fundamental challenge in machine learning: enabling agents with diverse objectives to cooperate effectively to achieve collective goals. The work described in this paper holds significant potential for various societal applications. By facilitating effective cooperation among agents with different motivations, our approach can be applied to domains such as video games, autonomous systems, smart cities, and decentralized resource allocation. These advancements can lead to numerous societal benefits, including improved efficiency in transportation systems, optimized resource allocation in energy grids, and enhanced decision-making in complex multi-agent environments. Furthermore, our findings contribute to the broader field of machine learning by offering insights into managing mixed-motivation scenarios. This work paves the way for future research and innovation, driving progress in multi-agent learning and its applications across various domains. However, while our method was initially developed to address fundamental challenges in decision-making within machine learning contexts, it is conceivable that our proposed objective alignment techniques could be repurposed for harmful purposes, such as attacking existing cooperative systems.

# B Proof of Corollary 4.3

**Corollary 4.3.** *In the neighborhood of fixed points of the collective objective, AgA will **pull** the gradient **toward** stable fixed points, which means $\theta(\tilde{\xi}, \nabla\mathcal{H}_c) \leq \theta(\xi_c, \nabla\mathcal{H}_c)$, and **push away** from unstable ones, indicated by $\theta(\tilde{\xi}, \nabla\mathcal{H}_c) \geq \theta(\xi_c, \nabla\mathcal{H}_c)$, if $\lambda$ satisfies $\lambda \cdot \langle\xi_c, \nabla\mathcal{H}_c\rangle(\langle\xi, \nabla\mathcal{H}_c\rangle + \|\nabla\mathcal{H}_c\|^2) \geq 0$.*

*Proof.* Prior to embarking on the elucidation of the corollary's proof, it is essential to first introduce relevant notations and the foundational concepts. Take note of the notation $\theta(a, b)$, which symbolizes the angular measure between two vectors. Furthermore, we write $\theta_\lambda(\tilde{\xi}, w)$ to represent the angular measure between AgA gradient $\tilde{\xi}$ and a reference update direction $w$. For simplicity, we use $\tilde{\xi} = u + \lambda v$ to denote the AgA gradient $\tilde{\xi} = \xi_c + \lambda\left(\xi + H_c^T\xi_c\right)$, as defined in Eq. 1. Specifically, $u$ denotes collective gradient $\xi_c$ and $v$ denotes $\left(\xi + H_c^T\xi_c\right)$.

We then extend the definition of infinitesimal alignment [Balduzzi et al., 2018] to our proposed AgA gradient. Given a third reference vector $w$, the ***infinitesimal alignment*** for AgA gradient $\tilde{\xi}$ with $w$ is defined as

$$align(\tilde{\xi}, w) := \frac{d}{d\lambda}\{cos^2\theta_\lambda\}_{|\lambda=0}. \tag{2}$$

Intuitively, we may presume that vectors $u$ and $w$ are directionally aligned, given that $u^Tw > 0$. In this scenario, $align > 0$ implies that vector $v$ is drawing $u$ closer towards the reference vector $w$. Conversely, vector $v$ is propelling $u$ away from $w$. Similar arguments hold for $align < 0$: $v$ pushes $u$ away from the reference vector $w$ when $u$ and $w$ share the same directional orientation. Conversely, $v$ pulls $u$ towards the reference vector $w$ when the vectors $u$ and $w$ are not oriented in the same direction.

The ensuing lemma provides a simplified method for determining the sign of $align$, circumventing the need for generating an exact solution.

**Lemma B.1** (Sign of $align$.)**.** *Given AgA gradient $\tilde{\xi}$ and reference vector $\nabla\mathcal{H}_c$, the sign of infinitesimal alignment $align$ could be calculated by*

$$sign\left(align(\tilde{\xi}, \nabla\mathcal{H}_c)\right) = sign\left(\langle\xi, \nabla\mathcal{H}_c\rangle(\langle\xi_c, \nabla\mathcal{H}_c\rangle + \|\nabla\mathcal{H}_c\|^2)\right).$$

*Proof.* By the definition of $\theta_\lambda$, we could directly get:

$$cos^2\theta_\lambda \tag{3a}$$

$$= \left( \frac{\langle \tilde{\boldsymbol{\xi}}, \nabla\mathcal{H}_c \rangle}{\|\tilde{\boldsymbol{\xi}}\|^2 \cdot \|\nabla\mathcal{H}_c\|^2} \right)^2 \tag{3b}$$

$$= \left( \frac{\langle \boldsymbol{\xi}_c + \lambda(\boldsymbol{\xi} + \nabla\mathcal{H}_c), \nabla\mathcal{H}_c \rangle}{\|\tilde{\boldsymbol{\xi}}\|^2 \cdot \|\nabla\mathcal{H}_c\|^2} \right)^2 \tag{3c}$$

$$= \frac{\langle \boldsymbol{\xi}_c, \nabla\mathcal{H}_c \rangle^2 + 2\lambda\langle \boldsymbol{\xi}_c, \nabla\mathcal{H}_c \rangle\langle \boldsymbol{\xi} + \nabla\mathcal{H}_c, \nabla\mathcal{H}_c \rangle + \mathcal{O}(\lambda^2)}{\left( \|\tilde{\boldsymbol{\xi}}\|^2 \cdot \|\nabla\mathcal{H}_c\|^2 \right)^2} \tag{3d}$$

$$= \frac{\langle \boldsymbol{\xi}_c, \nabla\mathcal{H}_c \rangle^2 + 2\lambda\langle \boldsymbol{\xi}, \nabla\mathcal{H}_c \rangle\langle \boldsymbol{\xi}_c, \nabla\mathcal{H}_c \rangle + 2\lambda\|\nabla\mathcal{H}_c\|^2\langle \boldsymbol{\xi}_c, \nabla\mathcal{H}_c \rangle + \mathcal{O}(\lambda^2)}{\left( \|\tilde{\boldsymbol{\xi}}\|^2 \cdot \|\nabla\mathcal{H}_c\|^2 \right)^2} \tag{3e}$$

$$= \frac{\langle \boldsymbol{\xi}_c, \nabla\mathcal{H}_c \rangle^2 + 2\lambda\langle \boldsymbol{\xi}_c, \nabla\mathcal{H}_c \rangle(\|\nabla\mathcal{H}_c\|^2 + \langle \boldsymbol{\xi}, \nabla\mathcal{H}_c \rangle) + \mathcal{O}(\lambda^2)}{\left( \|\tilde{\boldsymbol{\xi}}\|^2 \cdot \|\nabla\mathcal{H}_c\|^2 \right)^2}. \tag{3f}$$

In accordance with the established definition of infinitesimal alignment, it becomes feasible to further compute the sign of $align$ by taking the derivative of $\lambda$ with respect to $cos^2\theta_\lambda$:

$$sign\left( align(\tilde{\boldsymbol{\xi}}, \nabla\mathcal{H}_c) \right) \tag{4a}$$

$$= sign\left( \frac{d}{d\lambda} cos^2\theta_\lambda \right) \tag{4b}$$

$$= sign\left( \langle \boldsymbol{\xi}_c, \nabla\mathcal{H}_c \rangle \left( \langle \boldsymbol{\xi}, \nabla\mathcal{H}_c \rangle + \|\nabla\mathcal{H}_c\|^2 \right) \right). \tag{4c}$$

$\square$

Lemma B.1 empowers us to calculate the sign of infinitesimal alignment relying on components that are readily computable.

Let us observe $\langle \boldsymbol{\xi}_c, \nabla\mathcal{H}_c \rangle = \boldsymbol{\xi}_c^T \boldsymbol{H}_c \boldsymbol{\xi}_c$ for symmetric Hessian matrix $\boldsymbol{H}_c$. Under the assumption of $\boldsymbol{\xi}_c \neq 0$, we could derive:

$$\begin{cases} if\ \boldsymbol{H}_c \succeq 0, then\ \langle \boldsymbol{\xi}_c, \nabla\mathcal{H}_c \rangle \geq 0; \\ if\ \boldsymbol{H}_c \prec 0, then\ \langle \boldsymbol{\xi}_c, \nabla\mathcal{H}_c \rangle < 0. \end{cases} \tag{5}$$

As demonstrated by Eq. 5, when situated in the neighborhood of a stable fixed point (that is to say, $\boldsymbol{H}_c \succeq 0$), we observe $\langle \boldsymbol{\xi}_c, \nabla\mathcal{H}_c \rangle \geq 0$. In converse circumstances, $\langle \boldsymbol{\xi}_c, \nabla\mathcal{H}_c \rangle < 0$ occurs. Following this, we shall proceed to delve into two specific scenarios: stability and instability of the fixed point.

*1) Case 1: In a neighborhood of a stable fixed point.* If we are in a neighborhood of a stable fixed point then $\langle \boldsymbol{\xi}_c, \nabla\mathcal{H}_c \rangle \geq 0$. This indicates that both vectors $\boldsymbol{\xi}_c$ and $\nabla\mathcal{H}_c$ point in the same direction, i.e., $\theta(\boldsymbol{\xi}_c, \nabla\mathcal{H}_c) \leq \pi/2$. Referring to Lemma B.1, the sign of $align(\tilde{\boldsymbol{\xi}}, \nabla\mathcal{H}_c)$ is the same as the sign of $\langle \boldsymbol{\xi}, \nabla\mathcal{H}_c \rangle + \|\nabla\mathcal{H}_c\|^2$ if $\langle \boldsymbol{\xi}_c, \nabla\mathcal{H}_c \rangle \geq 0$. Consequently, we have $sign(\lambda) = sign(\langle \boldsymbol{\xi}, \nabla\mathcal{H}_c \rangle + \|\nabla\mathcal{H}_c\|^2) = sign(align)$ matching the claim of our corollary.

Let us now conjecture the scenarios of $sign(align)$, namely, $sign(align) \geq 0$ or $sign(align) < 0$. When $sign(align) \geq 0$, it follows that $sign(\lambda) \geq 0$. Consequently, vector $\boldsymbol{v}$ will draw $\boldsymbol{u}$ towards $\boldsymbol{w}$. Subsequently, when $sign(align) < 0$, we observe that $sign(\lambda) < 0$. A negative $\lambda$ reverses the mechanism such that $\boldsymbol{v}$ pushes $\boldsymbol{u}$ away from $\boldsymbol{w}$, aligning with the discussion following Eq. 2 that assumes a positive $\lambda$. Thus, regardless of the sign of $align$, if we set $sign(\lambda) = sign(\langle \boldsymbol{\xi}, \nabla\mathcal{H}_c \rangle + \|\nabla\mathcal{H}_c\|^2)$, vector $\boldsymbol{v}$ ensures $\boldsymbol{u}$ is drawn towards $\boldsymbol{w}$.

From now, we prove that in the neighborhood of a stable fixed point, if we let $sign(\lambda)$ satisfy $\lambda \cdot \langle \boldsymbol{\xi}_c, \nabla\mathcal{H}_c \rangle(\langle \boldsymbol{\xi}, \nabla\mathcal{H}_c \rangle + \|\nabla\mathcal{H}_c\|^2) \geq 0$, the AgA gradient $\tilde{\boldsymbol{\xi}}$ will be pulled more closer to $\nabla\mathcal{H}_c$ than $\boldsymbol{\xi}_c$.

*2) Case 2: In a neighborhood of a unstable fixed point.* The proofing approach for the unstable case exhibits similarity to Case 1.

Therefore, we prove that if we let the sign of $\lambda$ follows the condition $\lambda \cdot \langle \boldsymbol{\xi}_c, \nabla \mathcal{H}_c \rangle (\langle \boldsymbol{\xi}, \nabla \mathcal{H}_c \rangle + \|\nabla \mathcal{H}_c\|^2) \geq 0$, the optimization process using the AgA method exhibit following attributes. In areas close to fixed points, 1) if the point is stable, the AgA gradient will be pulled toward this point, which means that $\theta(\tilde{\boldsymbol{\xi}}, \nabla \mathcal{H}_c) \leq \theta(\boldsymbol{\xi}_c, \nabla \mathcal{H}_c)$; 2) if the point represents unstable equilibria, the AgA gradient will be pushed out of the point, indicating that $\theta(\tilde{\boldsymbol{\xi}}, \nabla \mathcal{H}_c) \geq \theta(\boldsymbol{\xi}_c, \nabla \mathcal{H}_c)$. An illustrative example of the corollary is provided in Fig. 2a.

$\square$

## C  Implementation of MARL Algorithms with AgA

The concrete implementation of altruistic gradient adjustment is elaborated in Algorithm 1. AgA can be easily incorporated into any centralized training decentralized execution (CTDE) framework of MARL. In a typical CTDE framework, each agent, denoted as $i$, strives to learn a policy that employs local observations to prompt a distribution over personal actions. The unique characteristic during the centralized learning phase is that agents gain supplementary information from their peers that is inaccessible during the execution stage. Furthermore, a centralized critic often operates in tandem in CTDE settings, leveraging shared information and evaluating individual agent policies' potency. To incorporate our proposed AgA algorithm into CTDE frameworks, supplementary reward exchanges between agents are required, offering the essential inputs for Algorithm 1. The individual losses are calculated from individual rewards conforming to established MARL algorithm framework, such as IPPO [de Witt et al., 2020] or MAPPO [Yu et al., 2022]. Here, the collective loss $\ell_c$ is calculated according to individual rewards; for example, it could be a sum of individual rewards or an elementary transformation thereof. Furthermore, the AgA algorithm requires a magnitude value of the alignment parameter $\lambda$, which should be a positive entity. The sign of $\lambda$ is then calculated in line 5 of Algorithm 1 and is based on each gradient component calculated in lines 2-4. Subsequently, the algorithm produces the adjusted gradient $\tilde{\boldsymbol{\xi}}$. $\tilde{\boldsymbol{\xi}}$ is then distributed across each corresponding parameter, and optimization is carried out using a suitable optimizer such as Adam optimizer [Kingma and Ba, 2015].

## D  Additional Experiment Results

### D.1  Result Visualization of Two-player Public Goods Game

Fig. 4 depicts the actions produced by various methods in a two-player public game. The action generated by each method is represented by a circle, which is color-coded according to the corresponding method. To ensure fairness, each method's updates are constrained to a maximum of 100 steps. These methods include the individual loss-driven simultaneous optimization (Simul-Ind), two variants of gradient adjustment methods termed as CGA and SGA, and the simultaneous optimization leveraging collective loss (Simul-Co) alongside our proposed AgA, where $\lambda = 1$. Each of these methods is fundamentally underpinned by the gradient ascent algorithm. The illustration consolidates data from 50 randomized runs initialized from diverse starting points. The 'X' demarcated circles indicate the average actions mapped to each method. From the figure, it's apparent that Simul-Ind, CGA, SGA, SVO tend to converge towards scenarios where at least one player abstains from contributing, an occurrence often tagged as 'free-riding'. In contrast, Simul-Co (represented by a yellow circle) and our AgA method (illustrated as a red circle) tend to converge towards scenarios characterized by

---

**Algorithm 1** Altruistic Gradient Adjustment (AgA)

1: **Input:** individual losses $\boldsymbol{\ell} = [\ell_1, \cdots, \ell_n]$, collective loss $\ell_c$, parameters $\boldsymbol{w} = [\boldsymbol{w}_1, \cdots, \boldsymbol{w}_n]$, magnitude of alignment parameter $\lambda$
2: $\boldsymbol{\xi} \leftarrow [grad(\ell_i, \boldsymbol{w}_i) \; for \; (\ell_i, \boldsymbol{w}_i) \in (\boldsymbol{\ell}, \boldsymbol{w})]$
3: $\boldsymbol{\xi}_c \leftarrow [grad(\ell_c, \boldsymbol{w}_i) \; for \; \boldsymbol{w}_i \in \boldsymbol{w}]$
4: $\nabla \mathcal{H}_c \leftarrow [grad(\frac{1}{2}\|\boldsymbol{\xi}_c\|^2, \boldsymbol{w}_i) \; for \; \boldsymbol{w}_i \in \boldsymbol{w}]$
5: $\lambda \leftarrow \lambda \times sign\left(\frac{1}{d}\langle \boldsymbol{\xi}_c, \nabla \mathcal{H}_c \rangle \left(\langle \boldsymbol{\xi}, \nabla \mathcal{H}_c \rangle + \|\nabla \mathcal{H}_c\|^2\right)\right)$
6: **Output:** $\tilde{\boldsymbol{\xi}} = \boldsymbol{\xi}_c + \lambda(\boldsymbol{\xi} + \nabla \mathcal{H}_c)$
{Plug into any gradient descent optimizer}

---

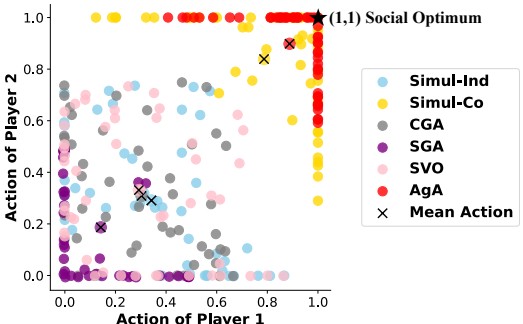

Figure 4: The scatter of actions in a two-player public goods game achieved through different optimization methods. Each circle represents the position attained within a maximum of 100 steps, with the color indicating the corresponding method. The 'X' mark represents the mean actions of 50 random runs. With the exception of Simul-Co, the baseline methods converge towards the Nash equilibrium (0,0). Notably, while both AgA and Simul-Co display altruistic behavior, the actions of AgA are more tightly clustered around the (1,1) point compared to Simul-Co.

enhanced contributions or 'altruism'. The observed results underline that players employing the AgA method exhibit heightened altruistic tendencies relative to those operating under Simul-Co. Further, AgA exhibits a superior alignment towards socially optimal outcome point $(1, 1)$.

## E    Experiment Details

### E.1    Sequential Social Dilemma Games: Cleanup and Harvest

SSD games [Vinitsky et al., 2019] implements the Harvest and Cleanup as grid world games. The agents use partially observed graphics observation, which contains a grid of $15 \times 15$ centered on themselves. Compared to the original Cleanup and Harvest games, SSD games add a fire beam mechanic, where agents can fire on the grid to hit other partners. An agent will gain 1 reward if harvesting an apple and -1 reward if firing a beam. Besides, being hit by a beam will result in a 50 individual reward loss. Under this mechanic, selfish agents can easily use fire to prevent others from harvesting apples to gain more individual rewards but harm social welfare.

We utilized PPO algorithm in stable-baselines3 [Raffin et al., 2021] to implement the baselines and our methods, with all the agents using separated policy parameters for Simul-Ind and sharing same policy parameters for other experiments. For SVO, we modify the individual reward to be $r_i - \alpha(1 - actan\left(\frac{sum_{j,j \neq i} r_j}{r_i}\right))$. To employ CGA and AgAs to PPO training process, we compute both individual and collective PPO-Clip policy losses and subsequently utilize them to calculate the adjusted gradient through automatic differentiation. We don't make changes to the critic loss nor the critic net optimization process.

Most experiments were conducted on a node with a Tesla V100 GPU (32GB memory) and 40 CPU cores. The hyper-parameters for PPO training are as follows.

- The learning rate is 1e-4
- The PPO clipping factor is 0.2.
- The value loss coefficient is 1.
- The entropy coefficient is 0.001.
- The $\gamma$ is 0.99.
- The total environment step is 1e7 for Harvest and 2e7 for Cleanup.
- The environment episode length is 1000.
- The grad clip is 40.

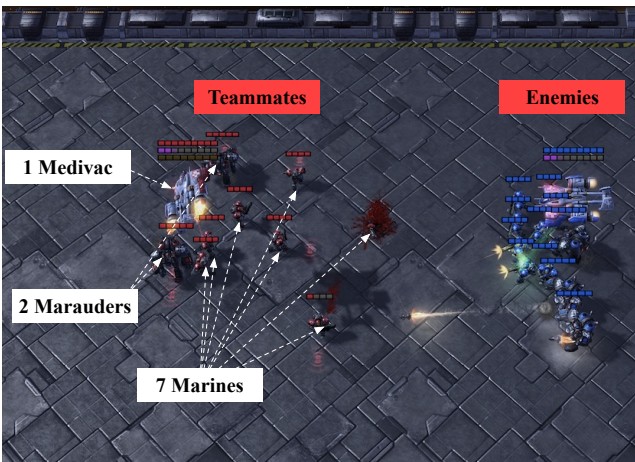

Figure 5: Semantic Diagram of MMM2 map in SMAC

## E.2 Selfish-MMM2

SMAC, often employed as a testbed within the sphere of cooperative MARL, is characterized by the shared reward system among players. Within this paper, we present the MMM2 map transformed as a versatile testbed designed for mixed-motive cooperation. Specifically, as shown in Fig. 5, MMM2 map encompasses 10 agents and 12 adversaries as enumerated below:

- Controlled Agents: Comprised of 1 Medivac, 2 Marauders, and 7 Marines.
- Adversaries: Incorporates 1 Medivac, 3 Marauders, and 8 Marines.

Moreover, this setting embodies both heterogeneity and asymmetry, where agents within the same team exhibit diverse gaming skills. Additionally, the team compositions, for instance, variation in agent number and type, differ in both opposing factions. Consequently, the intricate gaming mechanism, the presence of heterogeneous players, and the discernible asymmetry conspire to establish an exemplary milieu conducive for the exploration of mixed-motive issues.

In order to create a mixed-motive testbed, we adjusted the reward system hailing from the original SMAC environment. With this, we proposed a revamped reward function that features two distinct components: a reward for imposing damage and a penalty associated specifically with agent fatalities. In a move to potentially intensify internal conflicts amongst agents, the reward for imposing damage is solely allocated to the agent responsible for executing the attack on the enemy. Additionally, the death of an agent results in the imposition of an extra penalty. Consequently, this mechanism fosters an environment where agents are predisposed towards individual protection and separate reward acquisition, as opposed to collective cooperation.

Formally, the agent $i$'s reward at step $t$ is defined as follows:

$$\text{delta-enemy}_i = \sum_{\substack{j \in \text{Enemy} \\ i \text{ hit } j \text{ at } t}} \left[ (\text{previous-health} - \text{current-health}) + (\text{previous-shield} - \text{current-shield}) \right]$$

The individual reward for Player $i$ is determined as follows: If the player dies, then $r_i = \text{delta-enemy}_i - \beta \times \text{penalty}$; otherwise, if the player survives, then $r_i = \text{delta-enemy}_i$.

**Experiment settings.** Simul-Ind leverages IPPO with recurrent policies and distinct policy networks. Conversely, Simul-Co employs MAPPO with recurrent policies and consolidated policy networks. The implementation of the IPPO and MAPPO algorithms presented in this paper is founded on the methodology detailed in [Yu et al., 2022]. We utilize gradient adjustment optimization methods, such as CGA and AgAs, derived from MAPPO, implementing gradient adjustments as outlined in Algorithm 1.

Most experiments were conducted on a node with two NVIDIA GeForce RTX 3090 GPUs and 32 CPU cores. The hyper-parameters for PPO-based training are as follows.

- The learning rate is 5e-4
- The PPO clipping factor is 0.2.
- The value loss coefficient is 1.
- The entropy coefficient is 0.01.
- The $\gamma$ is 0.99.
- The total environment step is 1e7.
- The factor $\beta$ in reward function is 1.
- The environment episode length is 400.

