# OpenReview forum: "Aligning Individual and Collective Objectives in Multi-Agent Cooperation"
_NeurIPS.cc/2024/Conference — NeurIPS 2024 poster_

### Official Review · Reviewer_1xxK · 2024-06-30

**Soundness:** 3
**Presentation:** 3
**Contribution:** 3
**Rating:** 7
**Confidence:** 4

**Summary:**

The paper lies in the intersection of multi-agent and game theory. In particular, dealing with "mixed-motive cooperative games". This is, games in which the maximization of individual rewards hampers the maximization of collective rewards. This is, when agents seek to maximize their individual reward/utility, and disregard the collective utility they are worst off as a collective. These type of problems are usually called "social dilemmas" in game theory. The objective is to achieve the 'social optimum', this is, the maximum collective reward a group of agents can obtain, beyond the individual reward.

The methodology proposed in the paper is to align the individual rewards towards the collective maximization rewards via "gradient shaping". This is, move the gradients of the individual reward maximization policy towards the collective reward. In order to shift the individual rewards, the algorithm includes an "alignment" parameter that aligns two optimization problems (individual maximization vs collective optimization) that otherwise, will pull against each other.

Since the collective optimization point is an equilibrium point, the algorithm shows convergence.

**Strengths:**

1.Simplicity: The algorithm is simple and intuitive. It can be easily implemented in the current major multi-agent algorithms.

2.  Testing in well selected benchmark environments: The environments selected to test the algorithm are very standard in the field. Results show superiority against the selected benchmarks.

3. Ablation studies. The paper presents ablation studies for the relevant parameters. In particular, the "lambda" term controlling the alignment.

4. Presentation: the narrative is well written and makes the reading very straightforward. The paper provides intuition as well as rigor on the methodology proposed.

**Weaknesses:**

See below.

**Questions:**

I don't think this is fair or true. The mentioned papers do provide substantial theoretical analysis. This is not the way to place your paper.

"On the other hand, several studies focus on auto-matically modifying rewards by learning additional weights to adjust the original objectives [Gemp et al., 2020, Kwon et al., 2023]. However, these studies often suffer from a lack of interpretability and insufficient theoretical analysis regarding the alignment of individual and collective objectives."

**Limitations:**

Authors have addressed the limitations of the paper on the Conclusions section.

---

> ### Author Rebuttal · Authors · 2024-08-05
>
> **Question 1:** I don't think this is fair or true. The mentioned papers do provide substantial theoretical analysis. This is not the way to place your paper...
>
> **Response:** We apologize for the misunderstanding and appreciate your careful review. We have revised the statement as follows to make it more clear:
> “On the other hand, several studies focus on automatically modifying rewards by learning additional weights to adjust the original objectives [1,2]. Most of these methods leverage Nash equilibria or related concepts from game theory, such as the price of anarchy. The challenge of finding Nash equilibria in nonconvex games is more difficult than finding minima in neural networks [3], and it is not intuitive to explain how individual incentives align with the collective incentives during the optimization process."
>
> [1] I. Gemp, K. R.McKee, R. Everett, E. A. Duéñez-Guzmán, Y. Bachrach, D. Balduzzi, and A. Tacchetti. D3c: Reducing the price of anarchy in multi-agent learning. arXiv preprint arXiv:2010.00575, 2020.
> [2] M. Kwon, J. P. Agapiou, E. A. Duéñez-Guzmán, R. Elie, G. Piliouras, K. Bullard, and I. Gemp. Auto-aligning multiagent incentives with global objectives. In ICML Workshop on Localized Learning (LLW), 2023.
> [3] Letcher, A., Balduzzi, D., Racaniere, S., Martens, J., Foerster, J., Tuyls, K. and Graepel, T., 2019. Differentiable game mechanics. Journal of Machine Learning Research, 20(84), pp.1-40.

---

> > ### Comment · Reviewer_1xxK · 2024-08-07
> > **Thank you to the authors for their response.**
> >
> > Thank you for replying to the rebuttal and including my suggestion into your work. My rating stands.good luck with your paper!

---

> > > ### Author Response · Authors · 2024-08-10
> > > **Thanks**
> > >
> > > Dear Reviewer,
> > >
> > > Thank you for your encouraging message and for your valuable suggestions, which have significantly strengthened our work. Thanks~
> > >
> > > Best,
> > >
> > > All Authors

---

### Official Review · Reviewer_x8HM · 2024-07-01

**Soundness:** 2
**Presentation:** 3
**Contribution:** 3
**Rating:** 5
**Confidence:** 2

**Summary:**

The paper proposes Altruistic Gradient Adjustment (AgA) which adjusts the gradients of individual and collective losses to align individual and collective objectives. Besides, the authors prove that AgA effectively attracts gradients to stable fixed points of the collective objective while less sacrificing individual interests. Experiments and example illustrations demonstrate the effectiveness of AgA.

**Strengths:**

1.	The paper is well-written and well-organized.
2.	This topic is important with less attention in the MARL community.
3.	The experiments are performed on several environments and some example illustrations are given to help understand the proposed technique.
4.	The most important contribution of this paper is Corollary 4.3. This algorithm tries to seek stable fixed points for the collective objective while previous works aim to seek stable fixed points for individual objectives. This will possibly benefit the collective welfare as well as the individual interest at the same time, which would be useful as demonstrated in the experiments.

**Weaknesses:**

1.	AgA introduces an additional adjustment term, which needs to be tuned case by case.
2.	AgA introduces additional computation complexity to compute Hessian-vector products.
3.	It is not clear whether AgA also has the ability to always keep all individual objectives best. For example, in Figure 1(c), Simul-Co could achieve a better reward for Player 2 than AgA while Simul-Co also achieves a higher collective reward than AgA.
4.	There is only one map of SMAC to be tested.

**Questions:**

1. Could the authors compare the running time of AgA with other baselines?
2. Could the authors provide more discussion about AgA and Simul-Co? For example, if Simul-Co could achieve a higher collective reward than AgA like in Figure 1, one can redistribute the collective reward for each agent so that the redistributed individual rewards for each agent can be better than AgA.
3. In Algorithm 1, are the parameters w=[w_1,…,w_n] configured manually or determined by the task itself?

**Limitations:**

NA.

---

> ### Author Rebuttal · Authors · 2024-08-05
>
> **Weakness 1:** AgA introduces an additional adjustment term, which needs to be tuned case by case.
>
> **Response:**  Thank you for your feedback. Our AgA method indeed includes an gradient adjustment component, however, **the additional adjustment term is derived automatically based on our theoretical framework**, as outlined in Proposition 4.2 and Corollary 4.3, and is implemented in our derivation code. While the adjustment term itself does not require manual intervention, the parameter  $\lambda$  does need to be tuned for different tasks to ensure optimal performance.
>
> **Weakness 3:** It is not clear whether AgA also has the ability to always keep all individual objectives best. For example, in Figure 1(c), Simul-Co could achieve a better reward for Player 2 than AgA while Simul-Co also achieves a higher collective reward than AgA.
>
> **Response:** In mixed-motive settings, the group’s incentives can sometimes align and sometimes conflict [1][2]. Therefore, **it is almost impossible to consistently optimize all individual objectives simultaneously. If all individual objectives could always be optimal, the problem would reduce to a fully cooperative scenario where all incentives are perfectly aligned.** In Section 4.1, we emphasize the mixed-motive nature of our proposed differentiable mixed-motive game: "minimization of individual losses can result in a conflict between individuals or between individual and collective objectives (e.g., maximizing individual stats and winning the game are often conflict in basketball matches)."
>
> [1] McKee, Kevin R., Ian Gemp, Brian McWilliams, Edgar A. Duéñez-Guzmán, Edward Hughes, and Joel Z. Leibo. "Social diversity and social preferences in mixed-motive reinforcement learning." arXiv preprint arXiv:2002.02325 (2020).
> [2]Du, Yali, Joel Z. Leibo, Usman Islam, Richard Willis, and Peter Sunehag. "A review of cooperation in multi-agent learning." arXiv preprint arXiv:2312.05162 (2023).
>
> **Question 1:** Could the authors compare the running time of AgA with other baselines?
>
> **Response:** Please see common response.
>
> **Question 2:** Could the authors provide more discussion about AgA and Simul-Co? For example, if Simul-Co could achieve a higher collective reward than AgA like in Figure 1, one can redistribute the collective reward for each agent so that the redistributed individual rewards for each agent can be better than AgA.
>
> **Response:** Thank you for the insightful question regarding the comparison between AgA and Simul-Co. While Simul-Co focuses on maximizing collective rewards, it often overlooks individual agent incentives, potentially leading to dissatisfaction among agents with lower individual rewards. Redistributing collective rewards is also a potential method to address the problem. However, redistribution often encounters the credit assignment problem, one of the most challenging issues in the MARL field. It complicates the redistribution process, typically requiring expert manual design and substantial engineering effort.
> In this work, our goal is to propose an automatic method that modifies the gradient to seamlessly align individual and collective interests. Unlike traditional reward-shaping techniques, our approach doesn’t rely on manual intervention but instead automatically adjusts the gradient to ensure a balance between individual incentives and overall social welfare.
>
> **Question 3:** In Algorithm 1, are the parameters w=[w_1,…,w_n] configured manually or determined by the task itself?
>
> **Response:**  The parameters $w = [w_1, \dots, w_n]$ are initially defined in Definition 3 as follows: "The parameter set $w=[w_i]^n\in \mathbb{R}^d$ is defined, each with $w_i\in \mathbb{R}^{d_i}$ and $d = \sum_{i=1}^n d_i$. ... Each player $i\in N$ is equipped with a policy, parameterized by $w_i$, aiming to minimize its loss $\ell_i$." In practice, the parameters $w = [w_1, \dots, w_n]$ in Algorithm 1 refer to the neural network parameters, and our method is based on the popular PPO architecture. These parameters are learned and optimized during the training process rather than being manually configured.

---

> > ### Comment · Reviewer_x8HM · 2024-08-08
> >
> > Thank you for the detailed reply. Most of my questions and concerns are addressed. I would like to maintain my score.

---

> > > ### Author Response · Authors · 2024-08-10
> > > **Thanks**
> > >
> > > Dear Reviewer,
> > >
> > > Thank you for your feedback and for carefully reviewing our detailed reply.
> > >
> > > Best,
> > > All authors

---

### Official Review · Reviewer_sisd · 2024-07-11

**Soundness:** 4
**Presentation:** 4
**Contribution:** 4
**Rating:** 8
**Confidence:** 3

**Summary:**

This paper introduces a novel optimization method called AGA that employs gradient adjustments to progressively align individual and collective objectives. They prove that this method attracts gradients to stable fixed points of the collective objective while considering individual interests. Their method is empirically validated on sequential social dilemmas games, Cleanup and Harvest, and a high dimensional environment, StarCraft II.

**Strengths:**

- Preliminaries are clear
- Proofs are given graphs to explain the intuition
- Good comparison to baselines
- Varied experimental environments (grid game + modified SMAC)
- Strong experimental results

**Weaknesses:**

- No clear weaknesses

**Questions:**

- Figure 1. It’s not clear how to interpret the reward contours graphs. Can you clarify what the x-axis and y-axis? How are the contours formed? They appear to be integers corresponding to the Player’s actions but where is the action space defined?
- What is the significance of AGA moving along the summit?
- In Proposition 4.1, what is the gradient without the subscript referring to?
- L292, how is the collective loss equation determined?

**Limitations:**

Yes.

---

> ### Author Rebuttal · Authors · 2024-08-05
>
> **Question 1:** Figure 1. It’s not clear how to interpret the reward contours graphs...
>
> **Response:** Thank you for your advice to improve the clarity of the graph description. The x-axis and y-axis represent the actions of the two players, i.e., for player $i$ =\{1, 2\}, $a_i \in \mathbb{R}$, where $a_i$ is the action of player $i$. In our future version, we will include the definition of the action space in Example 4.1.  The contours in the graph represent the same rewards that can be achieved by the players based on different actions. The contours are formed by sampling the actions of the two players at regular intervals and calculating the corresponding rewards for each action pair. Points that yield the same reward are then connected to form the contour lines, representing levels of equal reward that can be achieved by the players based on their actions. Thank you again for your valuable advice. We will incorporate this explanation into the manuscript to make the graph clearer.
>
> **Question 2:** What is the significance of AGA moving along the summit?
>
> **Response:** According to the definition of Example 4.1, the reward function for Player 1 has a maximum value of 1. Thus, moving along the summit represents the algorithm's ability to maximize the player's reward during updates, ensuring that the player's interests are not overlooked. On the other hand, the Simul-Co method focuses solely on improving collective rewards and neglect individual player interests.
>
> **Question 3:** In Proposition 4.1, what is the gradient without the subscript referring to?
>
> **Response:** The gradient without the subscript is initially defined in Section 3.1 as follows: "We write the simultaneous gradient $\xi(w)$ of a differential game as $\xi(w) = (\nabla_{w_1}\ell_1, \dots, \nabla_{w_n}\ell_n) \in \mathbb{R}^d,$ which represents the gradient of the losses with respect to the parameters of the respective players." Since this definition is located far from the Proposition, we will reiterate it in our future version to provide clarity.
>
> **Question 4:** L292, how is the collective loss equation determined?
>
> **Response:** The collective loss function is determined experimentally within the context of mixed-motive problems and is inspired by the SVO method. Our motivation is to achieve equality while maximizing social welfare, ensuring that the loss function guides a fair contributions  among participants.

---

> > ### Comment · Reviewer_sisd · 2024-08-07
> >
> > Thank you for your response to my comments. I have read your rebuttal and will provide further comments soon, as needed.

---

> ### Author Response · Authors · 2024-08-10
> **Thanks**
>
> Dear Reviewer,
>
> Thanks for taking the time to review our response and your valuable input. We appreciate your continued engagement with our work.
>
> Best,
>
> All Authors

---

> > ### Comment · Reviewer_sisd · 2024-08-11
> >
> > I have no further comments and would like to maintain my score. Best of luck with the paper.

---

### Official Review · Reviewer_a1aR · 2024-07-12

**Soundness:** 3
**Presentation:** 1
**Contribution:** 2
**Rating:** 5
**Confidence:** 2

**Summary:**

The paper investigates the topic of cooperation in a mixed-motive multi-agent setting. They first propose the formulation of a mixed motive game as a differentiable game. Leveraging the structure of the latter, they propose a gradient-based optimization algorithm, AgA. The paper both discusses the theoretical guarantees of AgA, and its effectiveness in an empirical setting. Finally, the authors introduce a modification of the MMM2 map in the StarCraft II game, to further test the empirical performances of AgA.

**Strengths:**

- The paper is theoretically sound. The AgA algorithm is simple, but at the same time well justified.
- The experiment are broad and well designed. The Selfish-MMM2 environment addresses some of the limitations of the other ones, and I think is a principled way of showing the performance of the algorithm in larger and more complex environments

**Weaknesses:**

- There are some typos in the paper, and I feel the presentation can be improved. For example, in the Related Work section in page 2, there are some sentences which are incomplete or non-sensical. I am specifically referring to ' While PED-DQN enables agents 79 to incrementally adjust their reward functions for enhanced collaborative action through inter-agent 80 evaluative signal exchanges [Hostallero et al., 2020], Gifting directly reward other agents as part of 81 action space [Lupu and Precup, 2020].'
- I think the author should discuss more in depth the complexity of this method. How does it scale with respect to the other algorithms proposed? There is no experiment that highlights this in the paper
- There is no discussion of how AgA satisfied individual incentives

**Questions:**

- How much slower is AgA compared to the other algorithms? Do you have any empirical experiments which can shows this in practice?

- Do you have any results which show the individual metrics achieved by the agents when using AgA vs other algorithms? It could be valuable to see some plot which shows both the cooperative and individual rewards, and how these vary depending on the value the hyperparameter $\lambda$ is set to

- Did you run the experiments for Figure 3 for more seeds? Do the results still hold?

**Limitations:**

No limitations

---

> ### Author Rebuttal · Authors · 2024-08-05
>
> **Weakness 1: ... I am specifically referring to ' While PED-DQN enables age ...**
>
> **Response:** Thank you for your thorough and detailed reviews of my paper. We apologize for the unclear sentence and have rewritten it to improve clarity:
> *To further promote cooperation, the gifting mechanism—a crucial strategy in mixed-motive cooperation[2]—allows agents to influence each other’s reward functions through peer rewarding.
> Besides, PED-DQN[3] introduces an automatic reward-shaping MARL method that gradually adjusts rewards to shift agents’ actions from their perceived equilibrium towards more cooperative outcomes.*
>
> **Weakness 2 & Question 1: ...the complexity of this method... & How much slower is AgA compared to the other algorithm...**
>
> **Response:** Please see Common Response.
>
> **Weakness 3 & Question 2:There is no discussion of how AgA satisfied individual incentives & Do you have any results which show the individual metrics**
>
> **Response:**  Thank you for your valuable advice regarding individual metrics. Your suggestions are helpful and will contribute to improving the quality of our experiments and the overall paper. To evaluate individual incentives, we introduce the **Gini coefficient** [1], a commonly used measure of income equality, to assess reward equality in cooperative AI settings [4]. We utilized an expedited method for calculating the Gini coefficient and derived an associated equality metric $E$, defined as $E := 1 - G$. The Gini coefficient $G$ is computed from the ranked payoff vector $p$, which arranges each individual's rewards in ascending order, as $G = \frac{2}{n^2 \bar{p}} \sum_{i=1}^n i(p_i - \bar{p})$, where $\bar{p}$ is the mean of the ranked payoff vector $p$ and $n$ is the total number of players. *A higher $E$ value indicates greater equality.*
> **Table 1** presents the comparison of the mean and standard deviation of the equality metric achieved by different methods in the Harvest and Cleanup environments (note that the selfish-MMM2 environment involves heterogeneous agents, making it challenging to directly compare rewards achieved by different types of agents). A value closer to 1 indicates more equal rewards among agents. **As shown in Table 1, our proposed AgA method outperforms the baselines, demonstrating that AgA can effectively consider all interests within the team. Furthermore, as illustrated in Figure 3 of our manuscript, AgA achieves the highest collective rewards. Therefore, our AgA methods adequately address both individual and collective interests.**
>
> *Table 1: The comparison of the equality metric.*
> | **Envs**    | **Simul-Ind**             | **Simul-Co**              | **SVO**                   | **CGA**                   | **SL**                   | **AgA ($\lambda = 0.1$)**   | **AgA ($\lambda = 1$)** | **AgA ($\lambda = 100$)** | **AgA ($\lambda = 1000$)** |
> |-------------|---------------------------|---------------------------|---------------------------|---------------------------|---------------------------|------------------------------|-------------------------|---------------------------|-----------------------------|
> | **Harvest** | 0.973 ± 0.005             | 0.975 ± 0.006             | 0.974 ± 0.007             | 0.950 ± 0.051             | 0.972 ± 0.005             | 0.981 ± 0.006                | **0.988 ± 0.003**       | 0.982 ± 0.012             | 0.980 ± 0.006               |
> | **Cleanup** | 0.841 ± 0.071             | 0.948 ± 0.013             | 0.902 ± 0.019             | 0.903 ± 0.034             | 0.946 ± 0.016             | 0.940 ± 0.017                | 0.956 ± 0.007           | **0.959 ± 0.011**         | 0.905 ± 0.022               |
>
> **Question 3:** Did you run the experiments for Figure 3 for more seeds? Do the results still hold?
>
> **Response:** We run all algorithms with three seeds and report the mean and variance with a 95\% confidence interval. The results across different environments, including a toy game, a two-player public goods game, Harvest, Cleanup, and our developed selfish-MMM2 show that our method, AgA, consistently outperforms the baselines. Therefore, we believe that the results would remain consistent with more seeds.
>
> **Reference:**
>
> [1] David, H. A. Gini’s mean difference rediscovered. Biometrika, 55(3):573–575, 1968. ISSN 00063444. URL http://www.jstor.org/stable/2334264.
>
> [2] Lupu, A. and Precup, D. Gifting in multi-agent reinforcement learning. In Proceedings of the 19th International Conference on autonomous agents and multiagent systems, pp. 789–797, 2020.
>
> [3] D. E. Hostallero, D. Kim, S. Moon, K. Son, W. J. Kang, and Y. Yi. Inducing cooperation through reward reshaping based on peer evaluations in deep multi-agent reinforcement learning. In A. E. F. Seghrouchni, G. Sukthankar, B. An, and N. Yorke-Smith, editors, Proceedings of the 19th International Conference on Autonomous Agents and Multiagent Systems, AAMAS ’20, Auckland, New Zealand, May 9-13, 2020, pages 520–528. International Foundation for Autonomous Agents and Multiagent Systems, 2020. doi: 10.5555/3398761.3398825. URL https://dl.acm.org/doi/10.5555/3398761.3398825.
>
> [4] Du, Yali, Joel Z. Leibo, Usman Islam, Richard Willis, and Peter Sunehag. "A review of cooperation in multi-agent learning." arXiv preprint arXiv:2312.05162 (2023).

---

> > ### Comment · Reviewer_a1aR · 2024-08-08
> > **Main concerns addressed**
> >
> > Dear Authors,
> > Thanks for the answer and for the additional experiments provided. I suggest the authors to both include the discussion and results provided here on the individual incentive and the computational complexity of AgA in the final version of the manuscript.
> >
> > Since the main concerns I raised were addressed, I'll increase my score accordingly.

---

> > > ### Author Response · Authors · 2024-08-10
> > > **Thanks**
> > >
> > > Dear Reviewer,
> > >
> > > Thank you very much for your positive feedback and for the time you invested in reviewing our manuscript. We are pleased to hear that the additional experiments and the revisions we provided addressed your main concerns. **We will make sure to incorporate these elements to further enhance the clarity and completeness of our work.**
> > >
> > > Best,
> > >
> > > All Authors

---

### Author Rebuttal · Authors · 2024-08-04

Thank you to all the reviewers for your diligent efforts and invaluable feedback. Your comments will greatly contribute to enhancing the quality of our paper. We hope that we have addressed your concerns in our response. If you have any further questions, please don't hesitate to engage in discussion with us, and we will respond promptly.

**Common Response Regarding the Running Time of AgA:**

Thank you for the valuable comments regarding the running time and complexity of our proposed AgA algorithm. In this response, we first address the additional complexity introduced by AgA and then provide an analysis of its practical running time. First, the AgA method is a type of gradient adjustment method, where the modified gradient is given by $\xi_{c} + \lambda \left( \xi + H_c^T\xi_c \right)$ (see Proposition 4.2 in the paper). As discussed in the paper, it is not necessary to compute the Hessian matrix $H_c$ directly. Instead, we compute Hessian-vector products $H_c^T\xi_c$ for the modified gradient, which have a computational cost of $\mathcal{O}(n)$ for $n$ weights [1]. Thus, compared to most current methods in mixed-motive MARL, the additional complexity mainly arises from the Hessian-vector products. **As a result, the running time of AgA is generally expected to be more than twice as long as that of standard gradient-based methods.
The practical running time analysis also demonstrate the conclusion.** Table 1 presents the average running time of baseline methods and AgA in the two-player public goods game (see Sec. 5.1 in the paper). The table includes the total duration, total number of timesteps over 50 runs, the time per step, and the time per step ratio compared to our AgA method.
The experiments are conducted in Macbook Pro with Apple M1 Pro Chip.
The values in the ratio row are calculated by dividing the time per step of each method by the time per step of the AgA method, which facilitates an easy comparison of running times.
The Simul-Ind, Simul-Co, and SL are standard gradient-based methods, while CGA and SGA are gradient-modification methods. **Our findings indicate that the AgA method takes approximately 2-3 times longer per step compared to standard gradient-based methods. Additionally, AgA is slightly slower than other gradient-modification methods due to the more complex operations involved in sign judgment.** Despite having the highest running time, AgA is the most efficient method, requiring only 1389 steps for 50 runs, compared to around 4000 steps for the baselines.
Our experiments in Harvest, Cleanup, and Selfish-MMM2 were conducted on different servers, such as A100, V100, and 3090. Therefore, we cannot directly compare the running times across these setups. However, we have selected some experiments run on the same servers (A100 with 80 GB) to provide a fair comparison of the running times. In the Harvest environment, AgA takes approximately 100.33 minutes on average for training, compared to 47.33 minutes for Simul-Co. This difference, which is about a factor of two, is due to the acceleration provided by the A100 and PyTorch.

*We will incorporate the discussion into the future version. Thank you again to all reviewers for your valuable advice.*

*Table 1: Comparison of the running time between AgA and baseline methods.*
| Metrics | **Simul-Ind**             | **Simul-Co**              | **SL**                   | **CGA**                   | **SGA**                   | **AgA** |
|-------------|---------------------------|---------------------------|---------------------------|---------------------------|---------------------------|------------------------------|
Total Duration (ms)| 1165.79 | 910.15 | 1149.97 | 3041.84 | 3007.77 | 1034.69 |
Total Steps | 4272 | 3252 | 3887 | 4478 | 4179 | 1389
Time Per Step (ms)| 0.27 | 0.28 | 0.30 | 0.68 | 0.72 | 0.74 |
Ratio| 0.37 | 0.38 | 0.40 | 0.91 | 0.97 | 1.00 |

[1] B. A. Pearlmutter. Fast Exact Multiplication by the Hessian. Neural Computation, 6(1):147–160, 01 1994. ISSN 0899-7667. doi:10.1162/neco.1994.6.1.147.

---

### Decision · Program_Chairs · 2024-09-25

**Decision:**

Accept (poster)

**Comment:**

This work proposes a gradient adjustment technique for aligning individual and collective objectives in cooperative multiagent learning. The authors propose modifying the vanilla collective objective with a term that incorporates individual objectives as well as a consensus term that seeks fixed points of the collective objective. In experiments, this weight on the latter terms is set quite high such that the original collective objective plays less of a role.

The reviewers find the “The paper is theoretically sound. The AgA algorithm is simple, but at the same time well justified” with “Strong experimental results”. The reviewers were initially concerned about runtime, but the authors have argued hessian-vector products are efficient and reported only 2-3x slower runtimes empirically (please include these runtime results somewhere). The Gini coefficient is a nice addition to capture individual agent performance as requested by reviewers x8HM and a1aR. “The most important contribution of this paper is Corollary 4.3”, drawing inspiration from the analysis of SGA.

To improve the paper, it would be helpful to see more ablations on the three terms that make up the update. To align with individual objectives, it seems reasonable to assume the $\xi$ term is required. The question is then how to incorporate $\xi_c$ and $\nabla \mathcal{H}_c$.

Given the reviewers generally felt positive about the paper, we are recommending to accept this submission.

Regarding the [Gemp, Kwon] citations, I would remove the statement “it is not intuitive to explain how individual incentives align with the collective incentives during the optimization process.” In that work, individual vs collective alignment is precisely captured by a reward-sharing matrix during the learning process. One can intuitively measure alignment by measuring the norm of the difference between whatever the current sharing matrix is versus the collective sharing matrix (the ones matrix divided by number of agents).